Validation and modification of existing bleeding complications prediction models for percutaneous renal biopsy: a prospective study

Li Xing
Liu Min
Duan Di-fei
Yan Yu
Ma Dengyan mdy5104@wchscu.cn
Department of Nephrology, Kidney Research Institute, West China Hospital of Sichuan University/West China School of Nursing, Sichuan University , Chengdu , China
Capusa Cristina
Electronic publication date: 2024 Dec 18
Publication date: 2024
Volume: 12
Electronic Location ID: e18741
Received 2024 Sep 23; Accepted 2024 Nov 30
Copyright: © 2024 Li et al.
Copyright year: 2024
Copyright holder: Li et al.
License: This is an open access article distributed under the terms of the Creative Commons Attribution License, which permits unrestricted use, distribution, reproduction and adaptation in any medium and for any purpose provided that it is properly attributed. For attribution, the original author(s), title, publication source (PeerJ) and either DOI or URL of the article must be cited.
License URL: https://creativecommons.org/licenses/by/4.0/

Keywords: Percutaneous renal biopsy, Bleeding complications, Prediction model, Ultrasound, Decision curve analysis, Net reclassification improvement, Integrated discrimination improvement, Nomograms

Funding: West China Nursing Discipline Development Special Fund Project of Sichuan University HXHL21013 Scientific and Technological Department of Sichuan Province 2022YFS0266 This work was supported by the West China Nursing Discipline Development Special Fund Project of Sichuan University (HXHL21013), and the Scientific and Technological Department of Sichuan Province (No. 2022YFS0266). The funders had no role in study design, data collection and analysis, decision to publish, or preparation of the manuscript.

==============================
Background

Bleeding complications following percutaneous renal biopsy (PRB) are a significant clinical concern. This study aimed to validate and refine existing prediction models for post-biopsy bleeding to support more accurate clinical decision-making.

Methods

Clinical data from 471 PRB patients were examined in this prospective analysis. Ultrasounds were performed immediately and 6 h post-biopsy to identify perinephric hematomas. Patients exhibiting severe pain, a hemoglobin drop of >10 g/L, symptomatic hypotension, hematuria within 7 days post-procedure underwent repeat ultrasound to assess for bleeding complications. Univariate and multivariable logistic regression analyses were conducted to identify factors associated with bleeding risk. The predictive performance of three kidney biopsy risk calculators (KBRC) was evaluated using the area under the receiver operating characteristic (AUROC) curve, net reclassification improvement (NRI), integrated discrimination improvement (IDI), and decision curve analysis (DCA) to determine clinical utility. Nomograms were developed for each model to facilitate clinical application.

Results

Univariate analysis identified body mass index (BMI), hemoglobin, and ultrasound findings as significant predictors of bleeding complications. In multivariable analysis, BMI, immediate ultrasound, and 6-h ultrasound data remained significant (p < 0.05). The three models compared included: KBRC-5 (age, body mass index (BMI), platelet count, hemoglobin, kidney size), KBRC-5 with immediate ultrasound data (IKBRC), and KBRC-5 with 6-h hematoma size (SKBRC). The AUROC values for these models were 0.683, 0.786, and 0.867, respectively (p < 0.001). NRI and IDI analyses demonstrated that adding immediate or 6-h ultrasound data significantly improved the risk reclassification ability of the KBRC-5 model (p < 0.05). DCA indicated that IKBRC provided the highest net benefit for risk thresholds between 25% and 77%, while SKBRC was superior for thresholds between 10% and 95%. Nomograms were constructed for each model, allowing clinicians to estimate the probability of bleeding complications by summing scores for each predictor. Calibration curves showed good agreement between predicted and observed probabilities.

Conclusion

Incorporating real-time ultrasound data post-PRB significantly enhances the predictive accuracy and risk reclassification capability of bleeding risk models. These findings provide critical insights for guiding clinical management decisions in patients undergoing renal biopsy.

Introduction

Percutaneous renal biopsy (PRB) is a pivotal procedure in nephrology, instrumental in diagnosing the etiology and prognosis of renal diseases, guiding treatment decisions, and monitoring kidney transplant function and viability (Dhaun et al., 2014). Since its introduction roughly 70 years ago, PRB combined with histological analysis has become the gold standard in nephrology practice due to its ability to provide definitive insights into kidney pathology (Cameron & Hicks, 1997). However, despite its diagnostic significance, PRB is not without risks, with bleeding being the most frequent and concerning complication (Corapi et al., 2012). Bleeding complications occur in 11% to 37.7% of cases (Corapi et al., 2012; Li et al., 2024; Varnell, Stone & Welge, 2019), influenced by factors such as biopsy technique and patient characteristics. Severe bleeding can lead to prolonged hospital stays, increased treatment costs, and additional interventions (Hogan, Mocanu & Berns, 2016). In particular, the need for blood transfusions introduces immunological challenges, as it can increase sensitization and raise the risk of future immunologic complications for patients requiring kidney transplants (Lim et al., 2021). Given these risks, accurately predicting post-PRB bleeding is essential for patient management (Bhadauria et al., 2022). Effective risk prediction allows clinicians to tailor post-procedure care, including determining the appropriate length of bed rest and whether prophylactic treatments are necessary to prevent complications (Torigoe et al., 2022). Numerous studies have explored factors that may influence bleeding following PRB (Wang et al., 2022; Xu et al., 2021). Patient characteristics such as age, body mass index (BMI), and coagulation status have been consistently associated with bleeding risk (Li et al., 2024; Patel et al., 2021). Furthermore, impaired kidney function, low pre-biopsy hemoglobin levels, and elevated clotting times—such as activated partial thromboplastin time (aPTT) or prothrombin time (PT)—have also been identified as significant predictors of bleeding complications (Pombas et al., 2020).

In 2020, Schorr et al. (2020) developed a bleeding risk calculator that offers clinicians a practical tool for estimating post-biopsy bleeding risk based on pre-procedural patient characteristics. This model integrates variables such as age, BMI, platelet count, hemoglobin concentration, and maximum kidney size to calculate the individualized risk of bleeding. While this model has shown good predictive performance, it has limitations, particularly in not accounting for procedural factors that may influence bleeding risk. These factors are challenging to assess comprehensively in a predictive model.

The use of real-time ultrasound guidance during PRB has improved procedural safety and reduced complication rates. Ultrasound guidance not only aids in precise needle placement but also allows for the immediate detection of periprocedural complications, such as hematomas or arteriovenous fistulas (Zhan & Lou, 2023). In fact, performing an ultrasound immediately after the PRB has become standard practice to detect these complications early, thereby enabling timely intervention if needed. In 2021, Granata et al. (2021) emphasized the importance of performing a follow-up ultrasound after the PRB, even before moving the patient, to check for any adverse events such as hematomas, which is the most common periprocedural complications. Immediate ultrasound findings, such as the presence of a perinephric hematoma, could enhance the predictive accuracy of post-PRB bleeding risk by providing real-time information on complications that may have already developed.

Recognizing the limitations of existing models and the potential benefit of incorporating post-procedural data, we hypothesized that integrating both pre-procedure patient characteristics and immediate and 6-h post-PRB ultrasound findings would enhance the accuracy of bleeding risk prediction. To test this hypothesis, we validated the existing KBRC-5 model and compared its performance to modified models incorporating ultrasound data. We employed decision curve analysis (DCA) to evaluate the clinical utility of each model by assessing net benefit (NB) across a range of risk thresholds. DCA provides a comprehensive measure of a model’s practical applicability, offering valuable insights for clinical decision-making (Liu et al., 2024).

This study aimed to quantify the improvement in predictive performance achieved by incorporating ultrasound findings into bleeding risk models. We hypothesized that models combining patient characteristics with real-time ultrasound findings would outperform those relying solely on pre-procedural factors. The models were assessed using the area under the receiver operating characteristic curve (AUROC) and DCA. Furthermore, the added value of ultrasound data was evaluated through net reclassification improvement (NRI) and integrated discrimination improvement (IDI).

Materials and Methods

Study population and design

This study adhered to the Transparent Reporting of a Multivariable Prediction Model for Prognosis or Diagnosis (TRIPOD) + AI guidelines (Collins et al., 2024). Ethical approval was granted by the Biomedical Ethics Review Committee of West China Hospital, Sichuan University (protocol code: 2022-832), and all participants provided written informed consent. We conducted a prospective study, enrolling all adult patients undergoing PRB between November 2023 and August 2024 at West China Hospital, Sichuan University.

Peri-procedure management, biopsy procedures and data collection

Peri-procedural management aimed to minimize bleeding risks by withholding anticoagulants, antiplatelet agents, and other medications affecting coagulation prior to the procedure. For patients with platelet counts < 100 × 109/L, hemoglobin levels < 60 g/L, or an international normalized ratio (INR) > 1.5, corrective measures were undertaken, and bloodwork was rechecked. If the thresholds were still unmet, the procedure was delayed or rescheduled. Baseline laboratory tests included hemoglobin, platelet count, INR, creatinine, and urea levels. Neither desmopressin acetate nor conjugated estrogen was routinely used to mitigate bleeding risk. Blood pressure was monitored on the day of the procedure, with biopsies postponed if systolic blood pressure exceeded 150 mmHg or diastolic pressure was greater than 90 mmHg until adequate control was achieved.

In total, four different nephrologists performed biopsies during our study period. They had been performing PRB for at least 8 years and had participated in more than 1,000 PRB procedures. PRB were performed by nephrologists under ultrasound guidance, using a BARD Monopty 16-gauge, 16 cm Disposable Core Biopsy Instrument (Bard Peripheral Vascular, Inc, Tempe, AZ, USA), a spring-loaded device that provides a 22 mm tissue core. For all PRB patients in this study, the lower pole of the kidney was the target for puncture. Immediately following the biopsy, an ultrasound scan of the kidney was performed to document any perirenal hematoma in real-time. Standard post-procedure care included maintaining the patient in a supine position for at least 24 h, with continuous electrocardiographic monitoring. Vital signs, including pulse rate and blood pressure, were recorded every 2 h. Additionally, a routine follow-up ultrasound was conducted 6 h after the PRB to assess for any delayed hematoma. If any patient experienced a hemoglobin drop of >10 g/L, symptomatic hypotension, hematuria, or persistent pain within 7 days, an urgent repeat ultrasound was performed to evaluate for bleeding complications.

Outcomes

Outcomes were classified as either minor or major complications. Minor complications included perinephric hematoma detected by urgent ultrasound screening, gross hematuria, or bleeding that did not require intervention. Major complications were defined as bleeding events necessitating transfusion, surgical intervention, or embolization of a bleeding vessel.

Patient characteristics

Based on prior studies, we collected a comprehensive set of baseline characteristics known to influence bleeding risk. These included age, platelet count, hemoglobin concentration, kidney size, BMI, coagulation status, comorbidities, and the number of needle passes during the biopsy procedure.

Statistical analyses

All statistical analyses were performed using R version 4.4.0, with significance set at a two-tailed p-value < 0.05. Continuous variables were summarized as means when the data were normally distributed, and medians were used for skewed distributions. Categorical variables were reported as numbers and percentages.

Univariate logistic regression was performed to identify factors associated with bleeding complications. The logistic regression model is expressed as:

log⁡it(P)=ln(P1−P)=β0+β1X1+β2X2+...+βkXk,

where P represents the probability of bleeding complications, β0 is the intercept, and βi are the regression coefficients for the predictors Xi. Variables found to be significant in univariate analysis were included in a multivariable logistic regression model to adjust for potential confounders and determine the independent contributions of each factor.

The AUROC was calculated to assess and compare the predictive performance of three kidney biopsy risk calculators: (1) the Kidney Biopsy Risk Calculator (KBRC-5), (2) KBRC-5 combined with immediate post-biopsy ultrasound findings (IKBRC), and (3) KBRC-5 combined with 6-h post-biopsy ultrasound findings (SKBRC). The AUROC was computed using the following integral:

AUROC=∫01TPR(FPR−1(t))dt,

where TPR (true positive rate) and FPR (false positive rate) are functions of the classification threshold t.

To evaluate the clinical utility of these models, DCA was employed to compare their NB across a range of risk thresholds for predicting bleeding complications within 1 week post-PRB. The NB for a given threshold p was calculated as:

NetBenefit=TruePositivesN−FalsePositivesN⋅p1−p,

where N is the total sample size, and p represents the threshold probability. The method described by Vickers & Elkin (2006) was used to quantify the improvement in decision-making when incorporating sonographic data into existing prediction models.

Additionally, nomograms were developed for all three models, integrating significant predictors and clinically relevant variables. These nomograms provide clinicians with a user-friendly tool to estimate individual bleeding risk by assigning point values to each variable and calculating total scores to predict the probability of bleeding complications. The calibration curves of the nomograms were constructed to assess their agreement between predicted and observed outcomes, using the following Brier score formula:

BrierScore=1N∑i=1N(fi−Oi)2,

where fi represents the predicted probability, and Oi is the observed outcome (0 or 1).

Incremental predictive values of adding immediate or 6-h ultrasound data to the KBRC-5 model were further evaluated using NRI and IDI.

NRI was calculated as:

NRI=[P(↑|event)−P(↓|event)]+[P(↓|non−event)−P(↑|non−event)],

where P(↑) and P↓ denote the proportions of individuals with increased or decreased risk predictions.

IDI was calculated as:

IDI=(Peventnew¯−Pnon−eventnew¯)−(Peventold¯−Pnon−eventold¯),

where pevent¯ and pnon−event¯ represent the mean predicted probabilities for events and non-events, respectively, under the new and old models.

Results

Clinical characteristics

During the study period, a total of 471 patients underwent PRB (mean age 42.60 years [SD 14.28]; mean BMI 24.24 [SD 3.75]). The incidence of minor and major bleeding complications was 236 (50.1%) and 6 (1.3%), respectively. Among these patients, 32 (6.8%) developed macroscopic hematuria, 204 (43.3%) experienced hematoma alone, and 23 (4.9%) had both macroscopic hematuria and hematoma. One patient (0.2%) required a blood transfusion, while five patients (1.1%) required embolization or surgical intervention. The baseline characteristics of the study cohort are detailed in Table 1. Patients who experienced major bleeding complications were found to have significantly lower BMI and pre-biopsy hemoglobin levels. Although not statistically significant, major bleeding was also more frequent among those with smaller kidney size, lower platelet counts, higher serum creatinine levels, and systolic blood pressure (BP) ≥ 160 mmHg.

Table 1 Characteristics of patients according to postbiopsy bleeding events.

Characteristics	Total (N = 471)	No bleeding complications (N = 235)	Minor bleeding events (N = 230)	Major bleeding events (N = 6)	p value	
Age (yra), mean (SDb)	42.60 (14.28)	43.63 (14.46)	41.83 (14.11)	31.83 (7.17)	0.118	
BMIc, mean (SDb)	24.24 (3.75)	25.23 (3.79)	23.22 (3.47)	24.30 (2.72)	<0.001	
Maximum size of the kidney, median (IQRd)	10.3 (9.8–10.9)	10.4 (9.8–11.1)	10.3 (9.8–10.9)	9.75 (9.5–10.2)	0.324	
Systolic BPe					0.836	
<140 (mmHg)	409 (86.8)	206 (87.7)	198 (86.1)	5 (83.3)		
140–150 (mmHg)	47 (10)	22 (9.4)	25 (10.9)	0 (0)		
151–160 (mmHg)	7 (1.5)	4 (1.7)	3 (1.3)	0 (0)		
>160 (mmHg)	8 (1.7)	3 (1.2)	4 (1.7)	1 (16.7)		
Diastolic BPe					0.851	
<80 (mmHg)	218 (46.3)	110 (46.8)	107 (46.5)	1 (16.7)		
80–90 (mmHg)	164 (34.8)	81 (34.5)	81 (35.2)	2 (33.3)		
91–100 (mmHg)	72 (15.3)	34 (14.5)	36 (15.7)	2 (33.3)		
>100 (mmHg)	17 (3.6)	10 (4.3)	6 (2.6)	1 (16.7)		
Creatinine (μmol/L), median (IQRd)	101.0 (74.0–149.0)	98.0 (74.0–145.0)	109.0 (74.0–147.25)	145.5 (115.3–361.8)	0.276	
Hemoglobin (g/L), mean (SDb)	130.09 (114.0–146.0)	136.13 (26.536)	124.0 (26.63)	127.3 (21.62)	<0.001	
Platelets (×109), mean (SDb)	228.30 (178.0–270.0)	233.3 (81.0)	223.6 (76.3)	212.3 (59.3)	0.168	
Number of passes, median (IQRd)	2.00 (2–3)	2 (2–3)	2 (2–3)	2 (2–3)	0.232	
Diabetes mellitus					0.065	
No	423 (89.8)	203 (86.4)	212 (92.2)	6 (100)		
Yes	48 (10.2)	32 (13.6)	18 (7.8)	0 (0)		
Hypertension					0.749	
No	250 (53.1)	123 (52.3)	126 (54.8)	1 (16.7)		
Yes	221 (46.9)	112 (47.7)	104 (45.2)	5 (83.3)		
Notes:

a Years.

b Standard deviation.

c Body mass index.

d Interquartile range.

e Blood pressure.

Correlation with bleeding complications

Univariate analysis revealed that BMI, hemoglobin levels, and findings from both immediate and 6-h ultrasound scans were significantly associated with bleeding complications (Table 2). In multivariable logistic regression analysis, BMI (OR = 0.918; 95% CI [0.857–0.984]), along with immediate (OR = 2.432; 95% CI [1.315–4.496]) and 6-h ultrasound data (OR = 1.942; 95% CI [1.675–2.251]) remained significant predictors of bleeding (p < 0.05).

Table 2 Logistic regression analyses results for risk factors of bleeding complications from percutaneous renal biopsy.

Characteristics	Univariable logistic regression analysis	Multivariable logistic regression analysis	
OR	p value	95% CI	OR	p value	95% CI	
Age	0.990	0.117	[0.997–1.003]				
BMIa	0.858	<0.001	[0.813–0.906]	0.918	0.016	[0.857–0.984]	
Maximum size of the kidney	3.275	0.279	[0.727–1.096]				
Systolic BPb	1.003	0.584	[0.992–1.015]				
Diastolic BPb	1.001	0.883	[0.986–1.017]				
Creatinine	1.001	0.584	[1–1.003]				
Hemoglobin	0.982	<0.001	[0.975–0.990]				
Platelets	0.998	0.169	[0.996–1.001]				
Number of passes	0.820	0.260	[0.580–1.158]				
Diabetes mellitus	0.564	0.068	[0.305–1.043]				
Hypertension	0.943	0.749	[0.656–1.354]				
Immediate ultrasound information	7.388	<0.001	[4.476–12.194]	2.432	0.005	[1.315–4.496]	
Hematoma size at 6 h	2.236	<0.001	[1.941–2.575]	1.942	<0.001	[1.675–2.251]	
Notes:

a Body mass index.

b Blood pressure.

Kidney biopsy risk calculator model for the prediction of bleeding complications

To evaluate the predictability of bleeding complications, the AUROC was calculated. Variables that yielded significant results in the multivariable analysis were included, alongside additional factors such as age, platelet count, hemoglobin levels, and kidney size on ultrasound. Although these additional factors did not reach statistical significance, they were included due to their potential clinical relevance in predicting bleeding risk.

Three models were developed and compared. The first model, termed KBRC-5, included the following variables: age, BMI, platelet count, hemoglobin, and kidney size on ultrasound. The second model, IKBRC, was an extension of KBRC-5 that added immediate ultrasound findings. The third model, SKBRC, also built upon KBRC-5 but included the size of the hematoma identified at the 6-h ultrasound. The AUROCs for the models were 0.683 (p < 0.001; 95% CI [0.635–0.731]) for KBRC-5, 0.786 (p < 0.001; 95% CI [0.744–0.827]) for IKBRC, and 0.867 (p < 0.001; 95% CI [0.833–0.902]) for SKBRC (Fig. 1).

Figure 1 Comparison of the area under the receiver operating characteristics curve (AUROC) between KBRC-5, IKBRC and SKBRC.

Comparison of the area under the receiver operating characteristics curve (AUROC) between KBRC-5, IKBRC and SKBRC. The AUROC of KBRC-5 0.683 (p < 0.001; 95% CI [0.635–0.731]). The AUROC of IKBRC was 0.786 (p < 0.001, 95% CI [0.744–0.827]). The AUROC of SKBRC was 0.867 (p < 0.001, 95% CI [0.833–0.902]). x = 1 − sensitivity, y = sensitivity.

Predictive nomogram and calibration curve for three models

Three nomograms combining significant predictors and clinically relevant variables were developed (Fig. 2). Each variable was assigned a score on a point scale, and by summing the total score, clinicians can reference the total point scale to draw a straight line down to determine the estimated probability of bleeding complications. Risk factors included immediate ultrasound hematoma, ultrasound hematoma size at 6 h. Protective factors included age, platelet count, hemoglobin concentration and BMI. In addition, calibration plots (Fig. 3) demonstrated good agreement between predicted and observed probabilities for all models, with Brier scores of 22.5 (KBRC-5), 18.9 (IKBRC), and 14.2 (SKBRC), confirming their reliability.

Figure 2 Nomograms of the KBRC-5 model, IKBRC model, and SKBRC model.

(A) Nomogram for KBRC-5. (B) Nomogram for IKBRC. (C) Nomogram for SKBRC. The nomogram is used to find the position of each variable on the corresponding axis. Firstly, you draw a vertical line for each of the variables of your patient to the points axis for the score of each variable; secondly, you sum up the scores of all valuables you read on the points scale to obtain total points; finally, you draw a vertical line from the total points axis to determine the risk of bleeding complication at the lower line of the nomogram.

Figure 3 Calibration plots of the KBRC-5 model, IKBRC model, and SKBRC model.

(A) This calibration plot evaluates the agreement between predicted probabilities and observed outcomes for bleeding complications following percutaneous renal biopsy using the KBRC-5 model. (B) This calibration plot evaluates the agreement between predicted probabilities and observed outcomes for bleeding complications following percutaneous renal biopsy using the IKBRC model. (C) This calibration plot evaluates the agreement between predicted probabilities and observed outcomes for bleeding complications following percutaneous renal biopsy using the SKBRC model. The solid diagonal line represents perfect calibration, where predicted probabilities perfectly match the observed probabilities. The blue dots represent the observed frequencies of bleeding events within deciles of predicted probabilities, while the vertical error bars indicate the 95% confidence intervals for the observed frequencies.

DCA for three models

DCA was conducted to assess the clinical utility of the three models (Fig. 4). For patients with a risk threshold between 25% and 77%, IKBRC provided greater NB than either treating all patients or treating none. Similarly, SKBRC showed higher NB when the risk threshold ranged between 10% and 95%. Both IKBRC and SKBRC, which incorporated ultrasound findings, outperformed KBRC-5 across all risk thresholds. This analysis underscores the improved clinical decision-making potential when incorporating real-time ultrasound data into bleeding risk prediction models following PRB.

Figure 4 DCA for the KBRC-5, IKBRC, and SKBRC predict bleeding complications in adults following renal biopsy.

The solid black line represents the net benefit of not investigating any patients, assuming no one would experience bleeding complications. The solid gray line indicates the net benefit of investigating all patients, assuming all would have bleeding complications. The solid red line shows the net benefit of investigating patients based on KBRC-5 results only. The solid blue line reflects the net benefit when using KBRC-5 results plus immediate ultrasound, and the solid green line shows the net benefit when adding a 6-h follow-up ultrasoun.

Incremental predictive value of ultrasound data

NRIs and IDIs were calculated to evaluate the risk reclassification capability of the models. The addition of immediate (p = 0.000) or 6-h ultrasound (p = 0.000) increased NRI and IDI in the KBRC-5 model (all p < 0.05; Table 3, Fig. 5), indicating that inclusion of immediate or 6-h ultrasound information improved the risk reclassification ability of the prediction model. For IKBRC the NRI was 0.69 (P(↑|event)=44.5%,P(↓|non−event)=90.2%). For SKBRC, the NRI was 1.30 (P(↑|event)=79.2%,P(↓|non−event)=85.9%). These results suggest that the 6-h ultrasound data provided better discrimination between events and non-events than the immediate ultrasound data.

Table 3 Performance metrics of multivariate models with and without ultrasound information to predict bleeding complications after PRB.

Model	Continuous NRI	IDI	
Estimate	p value	95% CI	Estimate	p value	95% CI	
KBRC-5 model	Reference			Reference			
KBRC-5 model with immediate ultrasound information	0.694	0.000	[0.559–0.840]	0.135	0.000	[0.105–0.166]	
KBRC-5 model with 6 h ultrasound information	1.304	0.000	[1.095–1.455]	0.316	0.000	[0.276–0.357]	
Note:

PRB, Percutaneous renal biopsy; NRI, net reclassification improvement; IDI, integrated discrimination improvement.

Figure 5 NRI scatter plots comparing the standard KBRC-5 model with the modified IKBRC and SKBRC models.

(A) NRI scatterplot comparing the standard KBRC-5 model with the improved IKBRC model. (B) NRI scatterplot comparing the standard KBRC-5 model with the improved SKBRC model. The two scatter plots illustrate the reclassification of cases and controls using the improved model vs the standard KBRC-5 model, respectively. The x-axis represents the predicted probabilities of post-biopsy bleeding complications from the standard model, while the y-axis represents those from the improved model. Each point corresponds to an individual patient, with red points indicating cases (patients with bleeding complications) and black points representing controls (patients without bleeding complications). The diagonal line indicates no change in predicted probabilities between the two models. Points above the diagonal represent individuals for whom the improved model assigned a higher predicted probability compared to the standard model, while points below the diagonal represent individuals with a lower probability.

For IKBRC, the IDI was 0.14, reflecting an improvement in mean predicted probabilities for events and a reduction for non-events. For SKBRC, the IDI was 0.32, indicating even greater enhancement in discrimination between events and non-events compared to IKBRC. These results confirm the incremental value of adding ultrasound findings to the KBRC-5 model, as both IDI values were statistically significant (p < 0.05).

Discussion

In this study, we prospectively collected and analyzed data on bleeding complications in 471 patients who underwent PRB. We assessed the impact of incorporating immediate and 6-h post-biopsy ultrasound data into bleeding risk prediction models. Our findings demonstrate that adding ultrasound data substantially enhanced the predictive accuracy and risk reclassification capacity of the models. Furthermore, we compared the clinical utility and predictive power of three models: a base model (KBRC-5) and two enhanced models incorporating ultrasound data (IKBRC and SKBRC). The results showed that models with ultrasound data not only achieved higher predictive accuracy but also provided a greater NB across specific risk thresholds, highlighting their potential for more precise and clinically useful bleeding risk assessment.

The results demonstrate a marked improvement in the risk reclassification capability of the KBRC-5 model when immediate or 6-h ultrasound data is integrated. This enhancement, measured through significant increases in the NRI and IDI, underscores the model’s enhanced precision in assigning patients to appropriate risk categories—an essential factor for effective clinical decision-making (Pencina et al., 2008). Specifically, the incorporation of ultrasound findings significantly boosts the model’s accuracy in identifying high-risk patients, enabling clinicians to prioritize these individuals for closer monitoring and timely intervention (Van Den Bosch et al., 2021). This allows clinicians to better target individuals requiring closer monitoring or timely intervention. This enhancement is particularly relevant in a clinical context where accurate stratification of bleeding risk can guide monitoring intensity and therapeutic decisions, ultimately contributing to better patient outcomes (Antonopoulos et al., 2022). These findings support the clinical value of integrating real-time imaging data into predictive models to refine risk stratification and inform personalized patient management strategies.

Our study’s models offer several advancements over existing bleeding risk prediction models for PRB, particularly in terms of predictive accuracy and AUROC values. The KBRC-5 model, which included pre-biopsy variables such as age, BMI, platelet count, hemoglobin, and kidney size on ultrasound, demonstrated moderate predictive accuracy with an AUROC of 0.683. This is consistent with the results of earlier studies, which reported similar performance metrics for the original KBRC-5 model (Schorr et al., 2020). By incorporating immediate post-biopsy ultrasound findings, the IKBRC model showed a significant improvement in predictive accuracy, with an AUROC of 0.786. This immediate ultrasound assessment provides critical information about the initial impact of the biopsy on kidney tissue, including the detection of small hematomas or other abnormalities before the patient is moved (Bhattacharya et al., 2024). The ability to assess these early complications in real time allows for a more nuanced prediction of subsequent bleeding events. The SKBRC model, which includes hematoma size identified at the 6-h ultrasound, further enhances predictive performance, achieving an AUROC of 0.867. This substantial improvement highlights the value of delayed imaging in capturing the full extent of post-biopsy changes, as hematomas may enlarge or become more apparent over time (Prasad et al., 2015). However, while the SKBRC model demonstrated the highest accuracy, it is worth noting that obtaining 6-h ultrasound results may not always be feasible in routine clinical practice due to logistical challenges. Therefore, the IKBRC model, which uses more readily available immediate ultrasound data, may offer a more practical solution for many healthcare settings. In comparison with prior models, the integration of real-time ultrasound data (immediate and 6-h post-biopsy) in the IKBRC and SKBRC models led to higher AUROC scores, indicating improved accuracy in identifying patients at risk of bleeding complications. This improvement highlights the added value of dynamic, real-time assessment, which many previous models lack.

DCA provides a clearer understanding of how predictive models can inform clinical decisions by evaluating their NB across different risk thresholds (Vickers & Holland, 2021). In our study, both ultrasound-enhanced models outperformed the original KBRC-5 model across all risk thresholds. The IKBRC model offered a positive NB for patients with a bleeding risk threshold between 25% and 77%. Similarly, the SKBRC model, with its inclusion of 6-h ultrasound data, showed an even broader range of NB, from 10% to 95%. For patients with a bleeding risk threshold between 10% and 95%, the SKBRC model provides guidance that balances benefit and risk more effectively than either treating all patients or treating none. This broad applicability makes SKBRC a valuable tool for individualized patient care, as it allows clinicians to tailor their monitoring and intervention strategies based on specific risk levels (Whittier, Sayeed & Korbet, 2016). It can be seen that the combination of real-time ultrasound data enables personalized patient management and has the potential to reduce the incidence of serious complications by facilitating early identification and intervention of high-risk patients.

Similar to previous studies (Shidham et al., 2005; Trajceska et al., 2019), our univariable logistic regression analysis showed that specific vital signs, age, laboratory test results, ultrasound information were associated with bleeding complications after PRB. However, in the multivariable logistic regression analysis, variables associated with bleeding complications were the BMI immediate ultrasound information, and hematoma size at 6 h. This reduction in significant predictors suggests that certain variables identified in univariate analysis may act as confounders or mediators. For instance, lower BMI and hemoglobin levels are likely indirect markers of underlying patient frailty (Huynh et al., 2022; Zhang et al., 2023), which might be more directly linked to an increased bleeding risk following PRB. These insights underscore the importance of refining predictive models to focus on the most robust indicators of risk.

The nomograms derived from each model, particularly from the SKBRC model, significantly enhance clinical applicability by providing a rapid, visual tool for assessing bleeding risk. To improve the usability of the SKBRC model specifically, we developed a nomogram that integrates key predictors—hematoma size at 6 h, BMI, hemoglobin levels, platelet count, kidney size, and age. Each predictor is assigned a weighted score, allowing clinicians to quickly calculate a patient’s overall bleeding risk by summing these values. This accessible tool facilitates efficient and informed decision-making in busy clinical environments, as it enables practitioners to easily assess the likelihood of bleeding complications based on individualized patient profiles. The prominence of hematoma size at 6 h as the most heavily weighted predictor within the SKBRC nomogram highlights the importance of delayed ultrasound imaging in capturing clinically significant post-biopsy changes. This finding aligns with a recent studie suggesting that delayed imaging offers valuable insights into evolving hematomas, which may not be fully evident in immediate post-biopsy scans (Montoya et al., 2021). Unlike previous models that focused on pre-biopsy and immediate post-biopsy factors, our nomogram underscores the added predictive value of later imaging data, which enhances accuracy in stratifying bleeding risk.

Additionally, the nomogram emphasizes the contribution of other variables such as BMI, hemoglobin, and platelet count, which add essential context for assessing overall risk. Studies by Palsson et al. (2020) and Hasegawa et al. (2022) similarly highlighted the role of these parameters in bleeding risk, but did not integrate them as intuitively into clinical tools. By translating complex statistical models into a user-friendly, accessible format, our nomogram bridges this gap, making advanced risk prediction easily implementable in clinical workflows.

Overall, the integration of real-time ultrasound findings in the SKBRC model offers a new standard for bleeding risk prediction following PRB. By enhancing predictive accuracy and supporting timely clinical interventions, this model has the potential to improve patient outcomes and optimize resource allocation in healthcare settings.

However, some limitations should be considered. First, this study was conducted at a single tertiary-care institution, which may limit the generalizability of our findings. External validation in larger, multi-center cohorts is needed to confirm the utility of these modified models across diverse populations. Additionally, patients initially expected to undergo PRB but later excluded for various reasons were not part of the analysis, which may introduce selection bias. Moreover, certain relevant variables—such as biopsy needle size, operator experience, and biopsy depth—were not included in our dataset, potentially limiting the comprehensiveness of the predictive models.

Despite these limitations, our study provides valuable insights into predicting bleeding complications after PRB. The integration of real-time ultrasound data into the IKBRC and SKBRC models represents a significant advancement in risk prediction, with potential to enhance patient safety and improve clinical outcomes. Future research should focus on validating these models in varied clinical settings and identifying additional risk factors that could further refine bleeding risk prediction, thereby enhancing the models’ utility in routine clinical practice.

Conclusions

This study demonstrates the value of ultrasound data as a predictor of bleeding risk following PRB. By incorporating ultrasound information after PRB, the predictive performance of the KBRC model for bleeding risk significantly improves. This enhanced model provides clinicians with a more accurate tool to guide patient management decisions, including the duration of supine positioning, monitoring requirements, and the appropriate use of hemostatic interventions. These findings underscore the importance of ultrasound imaging in observing hematomas post-PRB and highlight its potential to support clinical decision-making and facilitate integration into medical practice.

Supplemental Information

Supplemental Information 1 Raw data.

Additional Information and Declarations

Competing Interests

Author Contributions

Human Ethics

Data Availability

The authors declare that they have no competing interests.

Xing Li conceived and designed the experiments, analyzed the data, prepared figures and/or tables, and approved the final draft.

Min Liu conceived and designed the experiments, performed the experiments, analyzed the data, prepared figures and/or tables, authored or reviewed drafts of the article, and approved the final draft.

Di-fei Duan conceived and designed the experiments, analyzed the data, authored or reviewed drafts of the article, and approved the final draft.

Yu Yan performed the experiments, prepared figures and/or tables, and approved the final draft.

Dengyan Ma performed the experiments, authored or reviewed drafts of the article, and approved the final draft.

The following information was supplied relating to ethical approvals (i.e., approving body and any reference numbers):

The West China Hospital of Biomedical Ethics Review Committee granted Ethical approval to carry out the study within its facilities (Ethical Application Ref: 2022-832).

The following information was supplied regarding data availability:

The raw measurements are available in the Supplemental File.

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
