# Peer review of "Validation and modification of existing bleeding complications prediction models for percutaneous renal biopsy: a prospective study"

_PeerJ, doi:10.7717/peerj.18741_

## Round 0.1 · original submission · Major Revisions

More details should be provided for the used methods (design and statistics) and a reorganization of the manuscript would be advisable.

·

Basic reporting

Manuscript ID 106451v1
This paper is related to reviewing the manuscript titled " Validation and modification of existing bleeding complications prediction models for percutaneous renal biopsy: a prospective study"
This study aimed to validate and refine prediction models for post-biopsy bleeding complications in 471 PRB patients. The study found that BMI, hemoglobin, and ultrasound findings were significant predictors of bleeding complications. In multivariable analysis, BMI, immediate, and six-hour ultrasound data remained significant. The study also compared three Kidney Biopsy Risk Calculators (KBRC): KBRC-5, IKBRC, and SKBRC.
Firstly, Although the proposed study is successful, performance analysis, evaluation results, organization, presentation, content and results are poor of the paper. So, major revision given in the following items need to be performed.

Experimental design

1) Improve the conclusion section, enhance the manuscript to convey the purpose, objectives, method and major findings, especially results in the items of convenience, interest, comfort, enhancing student’s self-confidence and subjective initiative.
2) Use abbreviations after the first use in the text, in the abstract and throughout the paper.
3) Neither the mathematical nor algorithmic expressions of these methods are given in the paper text. The authors urgently need to find a solution to this issue, and the mathematical equations of the methods must be given in the paper.
4) What are the contributions of the authors in this study in terms of for percutaneous renal biopsy? It is essential to clarify this issue.
5) In addition, the proposed model should be compared with new methods, from the results except some figures (Only three figures available).
6) Performance analyses and results are very few and insufficient. Increasing the results and including more detailed analyses in the paper would increase the value and scope of this paper.
7) The interpretation of the results and the discussion section are insufficient. These sections should definitely be increased and improved.
8) The conclusion section really needs to be improved
9) The resolution of the figures giving the analysis results should be increased.
10) Clean the paper of English spelling and punctuation errors

Validity of the findings

As above

Additional comments

My decision is major revision. I would like to inform you that if all the requested items are not completed in this revision, my decision will be to reject the application in the second round. Otherwise, I do not see any harm in publishing the manuscript once the above revisions are made.

Best regards.

Reviewer 2 ·

Basic reporting

Dear Authors,

Thank you for the opportunity to review this important manuscript. This study addresses the clinically significant issue of predicting bleeding after renal biopsy. I believe that the predictive model developed by the authors could offer valuable insights for future clinical practice. However, I would like to suggest a few areas for improvement before considering acceptance.

Bleeding during renal biopsy may indeed be influenced by procedural factors. While the authors note that experienced physicians performed the biopsies, please specify the criteria used to define “experienced”? Additionally, could you clarify which part (lower pole?) of the kidney was targeted for biopsy?

Regarding the post-biopsy complications, the authors present perirenal hematoma as a complication; however, mild hematoma is often observed in nearly all cases after renal biopsy. Could you provide more detail on how this study defined hematoma as a complication?

In the multivariate analysis, what criteria were used for variable selection? Relying solely on factors with significant results in univariate analysis may be insufficient. Known factors associated with post-biopsy complications, as reported in the literature, should also be considered.

Experimental design

Please see the Basic reporting.

Validity of the findings

Please see the Basic reporting.

Additional comments

Please see the Basic reporting.

---

## Round 0.2 · accepted · Accept

All the reviewers' suggestions were resolved by the authors.

Reviewer 2 ·

Basic reporting

To the authors,

Thank you for the opportunity to review the revised manuscript. The comments raised by the reviewer are well addressed.

Experimental design

Please see the basic reporting.

Validity of the findings

Please see the basic reporting.

Additional comments

Please see the basic reporting.